# Influence of Guardrails on Track–Bridge Interaction with a Longitudinal Resistance Test of the Fastener

**Kaize Xie** [1,*], **Weiwu Dai** [1], **Hao Xu** [2] and **Weigang Zhao** [1,*]

[1] Key Laboratory of Railway Industry of Infrastructure Safety and Emergency Response, Shijiazhuang Tiedao University, Shijiazhuang 050043, China

[2] School of Civil and Engineering Management, Guangzhou Maritime University, Guangzhou 510725, China

[*] Correspondence: kzxie1988@stdu.edu.cn (K.X.); zhaoweig2002@163.com (W.Z.);
Tel.: +86-17-3679-18857 (K.X.); +86-13-6133-10073 (W.Z.)

**Featured Application: (1) The test results provide parameters for the standardized design of continuous welded rail on bridges. (2) The integrated simulation model and the recommended maximum value of continuous girder bridges can simplify the design process of continuous welded rail, and suggestions on the installation torque of the guardrail fastener and guardrail joint can provide guidance for the maintenance of continuous welded rail tracks. (3) The results give a new basis and reference for improving industry specifications for the design and maintenance of railway continuous welded rail.**

**Abstract:** The guardrail is an indispensable part of ballasted track structures on bridges. In order to reveal its influence on the track–bridge interaction of continuous welded rail (CWR), the longitudinal resistance model of the guardrail fastener and its influential factors are established through tests. By taking a continuous girder bridge (CGB) for railways as an example, a stock rail-guardrail-sleeper-bridge-pier integrated simulation model is developed. The effects of the guardrails, installation torque of the guardrail fastener, and joint resistance of the guardrail under typical conditions are carefully examined. The research results indicate that the nominal longitudinal resistance of the guardrail fastener and the elastic longitudinal displacement of the rail prior to sliding approximately grow linearly with the growth of the installation torque. The presence of a guardrail can alleviate the track–bridge interaction in the range of the CGB, but exacerbate the interaction near the abutment with moveable bearings. This fact enables the abutment position to be considered as a new control point for the design of CWR on bridges. Considering the changing rules of the rail longitudinal force and rail gap, it is recommended that the installation torques of the guardrail fastener and guardrail joint are 40–60 N·m and 700–800 N·m, respectively. The recommended maximum longitudinal stiffness of piers for CGBs is evaluated. When the longitudinal stiffness of the pier for a CGB is lower than the recommended value, the influence of the guardrail can be neglected in the design of the CWR.

**Keywords:** continuous welded rail; track–bridge interaction; guardrail; longitudinal resistance of fastener; joint resistance; longitudinal stiffness of the pier

## 1. Introduction

Train derailment not only causes economic losses, but also leads to the loss of life [1,2]. Preventing train derailment has always been pursuit of researchers [3–7]. Guardrails are commonly employed to avoid wheel derailment, as well as the derailed train hitting the bridge or falling from the bridge. Furthermore, they have become a crucial part of the ballasted track structure on the bridge.

Many researchers have performed numerical simulations and experiments to examine the functioning principle of guardrails and optimize the profile, as well as the installation

parameters, of guardrails. Ongoing research has resulted in a continuous strengthening of the function of the guardrails [8–14]. Due to the connection between the guardrails and the stock rails through the sleeper and fastener, guardrails laying on the bridge can undertake the load of the CWR together with the stock rail, thereby affecting the mechanical behavior of the latter. In a feasibility study of laying 50 m rails on simply supported bridges, Qiu et al. [15] first proved that the existence of guardrails and the longitudinal resistance of guardrail fasteners could affect the mechanical behavior of the 50 m stock rails. Additionally, they suggested that the guardrails should be considered in the reserved joint gap design of the 50 m stock rails on the bridge. Wei et al. [16] showed that the guardrails affect the stability of the CWR, and only when the temperature force aggregated in the guardrails is small, the guardrails could improve the stability of the CWR. As a result, these studies confirm that the presence of an appropriate guardrail system on the bridge also influences the track–bridge interaction of the CWR on the bridge.

The study on track–bridge interaction essentially originates from the calculation of the distribution of train braking and traction force in rails and bridges [17]. Subsequently, scholars across the globe have performed pioneering research by developing calculation models, track–bridge longitudinal resistance models, calculation methods [18–20], and load combinations. The track structure models range from traditional ballast tracks to various types of ballastless tracks [21–24]. Additionally, there is a wide range of existing bridge models, from reinforced concrete bridges to long-span flexible steel bridges with a highly nonlinear geometry to curved bridges [25–28]. The overall developed models of the CWR on bridges range from the truss model to the rail-beam-pier integrated model and the spatial solid model [29–31]. The calculation methods cover a broad range of approaches, such as the transfer matrix method and the Runge–Kutta method [32,33]. Moreover, a solution algorithm for the corresponding nonlinear equation, including the generalized variational principle and the finite element method, has been widely utilized [34,35]. The track–bridge longitudinal resistance models include the constant resistance model, the nonlinear resistance model accounting for the load history, and the resistance model considering the track frame effect [36–38]. The applied loads studied also cover single temperature load or train load, as well as various load combinations and special loads such as seismic load [39,40]. However, the influence of guardrails on the track–bridge interaction has been ignored in these studies, and the impact on the reliability and safety of the CWR on the bridge is incalculable.

Based on the above literature survey, this paper aims to carry out longitudinal resistance tests of the guardrail fastener and proposes a stock rail-guardrail-sleeper-beam-pier integrated simulation model to scrutinize the guardrail effect on the track–bridge interaction rules of the CWR. Further, the obtained rules can provide a solid basis and an appropriate reference for the refined design and maintenance of the CWR on the bridge.

## 2. Longitudinal Resistance Test of the Guardrail Fastener

### 2.1. Test Process

Based on the longitudinal resistance test principle of the fastener in Ref. [41], the longitudinal resistance test of the 50 kg/m guardrail fastener, as shown in Figure 1, is methodically carried out. Firstly, the guardrail fastener is correctly installed, and the installation torque is checked with a torque wrench. Secondly, the load is directly exerted on the center of the rail bottom employing a hydraulic jack, and the applied load is measured with a pressure transducer. Finally, the rail longitudinal displacement subjected to the load is appropriately tested via a displacement sensor. The applied load on the rail and the longitudinal displacement of the rail are collected by utilizing a dynamic signal acquisition as the sampling frequency is set equal to 100 Hz.

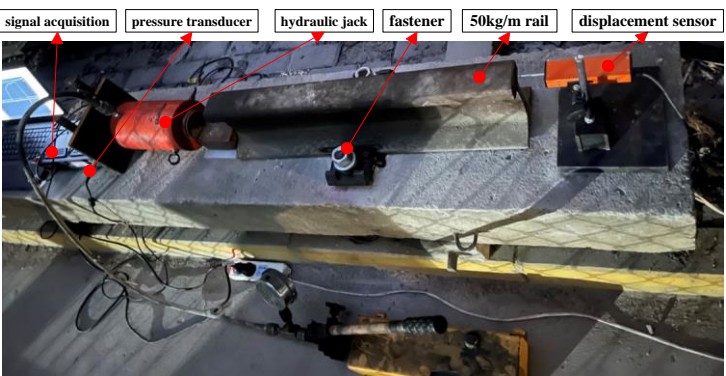

**Figure 1.** Longitudinal resistance test of the guardrail fastener.

*2.2. Test Results*

Based on the installation torque of the guardrail fastener, five test conditions with installation torques of 20, 40, 60, 80, and 100 N·m are set. Each test condition is successfully loaded four times, and the last three data sets are employed for analysis. Figure 2 illustrates a load–displacement curve with an installation torque of 60 N·m.

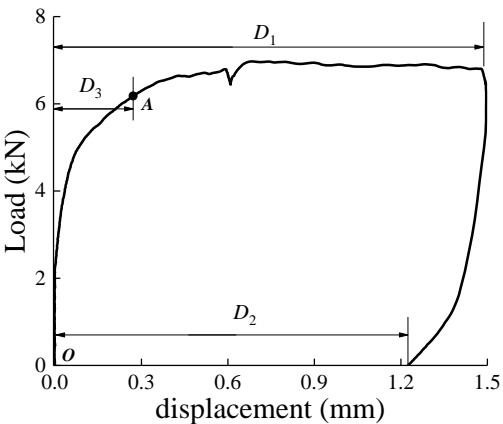

**Figure 2.** Load–displacement curve.

In Figure 2, $D_1$ represents the maximum longitudinal displacement of the rail, $D_2$ denotes the residual longitudinal displacement of the rail after the removal of load, and $D_3$ is the elastic longitudinal displacement of the rail prior to slipping. Furthermore, the load corresponding to $D_3$ is defined as the nominal longitudinal resistance of the fastener ($F_0$). During the design of CWR on the bridge, the rising phase of the load–displacement curve of the fastener (OA branch, as depicted in Figure 2) is often simplified to a straight line, and the longitudinal resistance of the fastener is simplified to a bilinear model. Due to the specific characteristics of the guardrail fastener, a huge error will be generated as a straight line is used to simulate the OA branch. Hence, the Langmuir EXT1 function is introduced to simulate the OA branch. The piecewise function, which is utilized to simulate the load–displacement curve of the guardrail fastener, can be expressed as follows:

$$F = \begin{cases} \frac{abx^{1-c}}{1+bx^{1-c}} & x < D_3 \\ F_0 & x \geq D_3 \end{cases} , \tag{1}$$

where $a$, $b$, and $c$ are constants to be determined, while the values of $D_3$ and $F_0$ are evaluated by utilizing three load–displacement curves.

Figure 3 demonstrates the fitting results of three rising phases of the load–displacement curves when the installation torque of the fastener is 60 N·m.

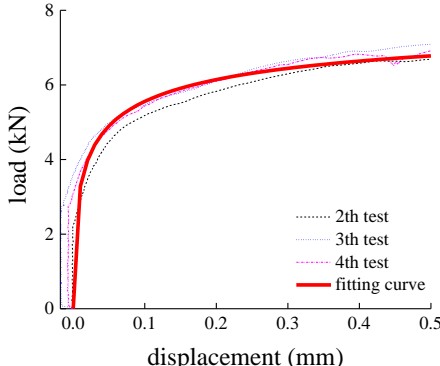

**Figure 3.** Fitting results (60 N·m).

Table 1 presents the fitting results associated with various installation torques, and Figure 4 illustrates the fitting curves.

**Table 1.** Summary of fitting results.

| Torque (N·m) | $A$ | $b$ | $c$ | $R^2$ | $F_0$ (kN) | $D_3$ (mm) |
|---|---|---|---|---|---|---|
| 20 | 3.67 | 10.06 | 0.52 | 0.8651 | 2.69 | 0.07 |
| 40 | 5.26 | 8.18 | 0.49 | 0.9530 | 3.91 | 0.15 |
| 60 | 8.29 | 6.32 | 0.51 | 0.9388 | 6.46 | 0.29 |
| 80 | 9.97 | 9.14 | 0.40 | 0.9718 | 8.39 | 0.48 |
| 100 | 14.03 | 2.88 | 0.55 | 0.9726 | 9.97 | 0.66 |

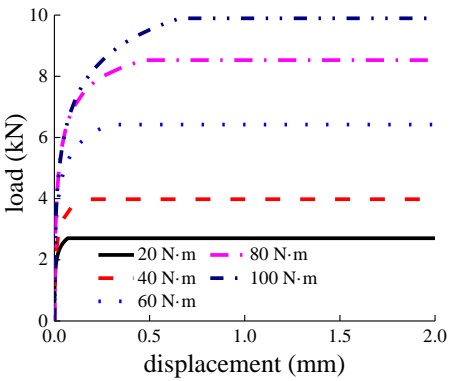

**Figure 4.** Fitting results for various installation torques.

Table 1 and Figure 4 display that the values of $D_3$ and $F_0$ with the installation torque are most relevant, which are approximately linearly correlated, as presented in Figure 5. In the figure, the parameter $M$ represents the installation torque of the guardrail fastener.

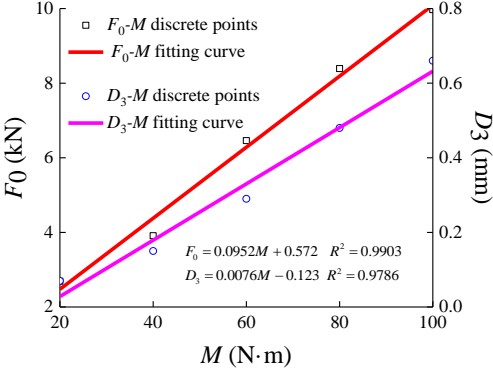

**Figure 5.** Relationship between the factors $F_0$, $D_3$, and $M$.

## 3. Materials and Methods

### 3.1. Model Establishment and Verification

In the existing calculation models of track–bridge interaction, the influence of the guardrail has not been taken into account. As the longitudinal resistance of the stock rail fastener is greater than that of the ballast bed, the sleeper is each time ignored and the double-layer elastic beam model is often adopted [21,22]. However, the guardrail longitudinal force can be delivered to the stock rail through fasteners of the guardrail and stock rail when the sleeper moves. The connections between various structures, including the sleeper, the stock rail, and the guardrail should be appropriately considered in the simulation model. Additionally, the stock rail-guardrail-sleeper-bridge-pier integrated simulation model demonstrated in Figure 6 is established. In this figure, the LRSRF and LRGRF in order represent the longitudinal resistance of the stock rail and guardrail fastener.

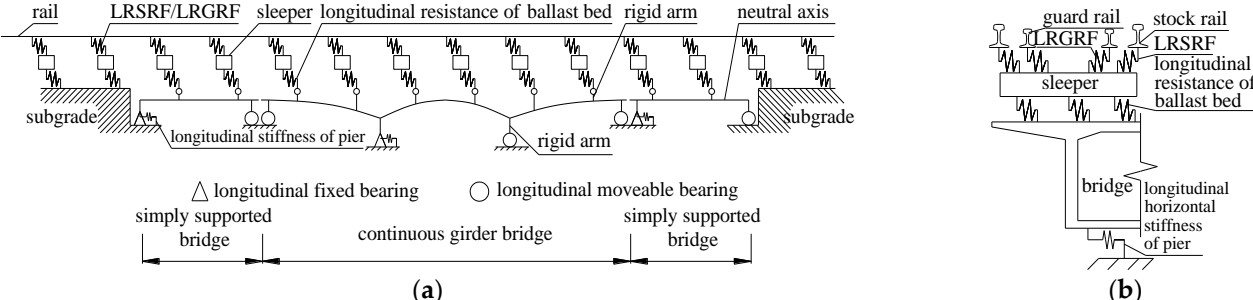

**Figure 6.** Stock rail-guardrail-sleeper-beam-pier integrated simulation model. (**a**) Profile view; (**b**) Cross-section view.

Different parts of the model are simulated with various types of elements according to their mechanical characteristics. Additionally, stock rails, guardrails, and sleepers are simulated by employing the Timoshenko beam element. The bridge, including the CGB and the simply supported main bridge structure, are simulated by combining the beam element and the rigid arm element. The beam element contains the cross-sectional characteristics of bridges, and the rigid arm element characterizes the bending deformation of the upper and lower flanges with reference to the neutral axis [42]. The fasteners connecting the stock rail and guardrail to sleepers, and the ballast bed connecting the sleepers to the upper of bridges are simulated with nonlinear spring elements. The piers and abutments are simplified by using linear spring elements with a different longitudinal stiffness. In order to reduce the influence of the boundary conditions on the calculation results, a subgrade with a length of more than 150 m is taken into account at both sides of the left and right abutments.

Considering the CWR on a (32 + 48 + 32) m CGB for the ballast track introduced in Ref. [22] as an example, the comparison results from the reference paper and the calculation mode shown in Figure 6 are presented in Table 2. In order to eliminate the influences of various models, the LRGRF (LRSRF) is taken as the infinitely small (large) value.

**Table 2.** Comparison results of different models.

| Torque (N·m) | Stock Rail Longitudinal Force Amplitude (kN) | | Pier Longitudinal Force Amplitude (kN) | | Stock Rail Displacement Amplitude (mm) | |
|---|---|---|---|---|---|---|
| | Ref. [22] | Present Study | Ref. [22] | Present Study | Ref. [22] | Present Study |
| Expansion | 209.2 | 206.2 | 126.4 | 122.8 | 6.4 | 6.3 |
| Bending | 30.2 | 35.4 | 7.0 | 7.0 | 0.25 | 0.28 |
| Braking | 358.0 | 363.6 | 264.2 | 269.9 | 11.2 | 11.5 |

Table 2 presents that the provided calculation results of Ref. [22] and those of the present study are basically consistent. This indicates that the model established could accurately simulate the track–bridge interaction of CWR on bridges. By setting the LRGRF

and LRSRF to normal values, the influence of guardrails on the track–bridge interaction can be analyzed.

### 3.2. Parameters of Bridges

A double-track railway reinforced concrete CGB with a span of (50 + 92 + 50) m is taken as an example. The cross-sectional parameters of the bridge are demonstrated in Figure 7. The abscissa of the left and right beam-end is specified by 0 m and 193.5 m, respectively. Five-span 32 m reinforced concrete simply supported bridges are laid on the left and right side of the CGB as boundary conditions. The cross-sectional area, vertical moment of inertia, beam height, and upper flange height of the simply supported bridges are 8.52 m$^2$, 9.35 m$^4$, 2.83 m, and 1.05 m, respectively.

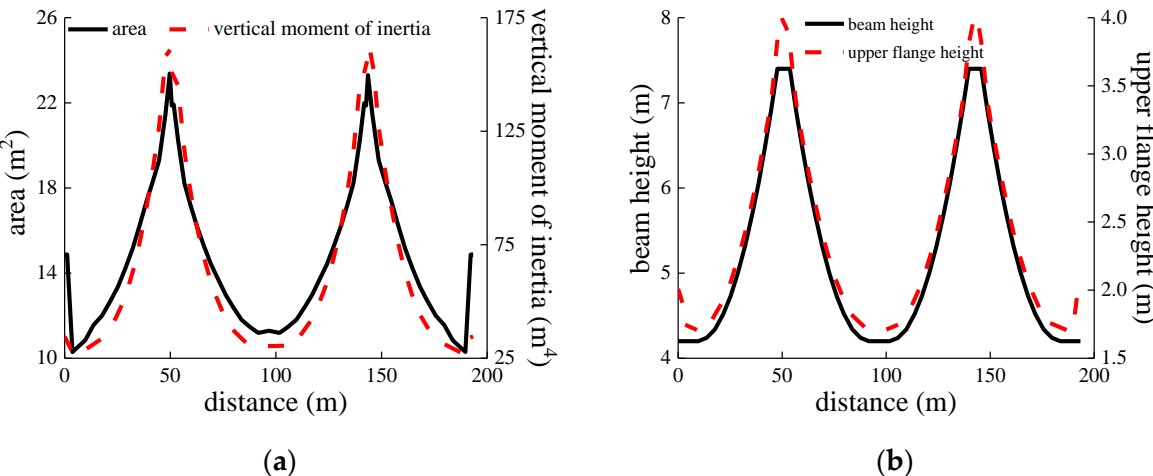

**Figure 7.** Cross-sectional parameters. (**a**) Area and vertical moments of inertia; (**b**) beam and upper flange height.

The longitudinal stiffness of the pier for CGB with fixed bearing is set as 1500 kN/cm. The longitudinal stiffness of abutments and piers for simply supported bridges are set equal to 3000 kN/cm and 350 kN/cm, respectively. The amplitude of the temperature variation of the consisting bridges is 15 °C [43].

### 3.3. Parameters of the Track

The mass per unit length of the stock rail and guardrail are considered as 60 kg and 50 kg, respectively. The length of each guardrail is set equal to 25 m, so the total numbers of guardrails and guardrail joints are, respectively, 88 and 84 for the double-track railway in the calculation model. The distance between the guardrail shuttle head and its adjacent abutment is set as 18 m. A new type-III bridge sleeper is utilized and the sleeper spacing is set equal to 60 cm. The cross-sectional area, the longitudinal moment of inertia, the height, and the length of the sleeper are $6.78 \times 10^{-2}$ m$^2$, $4.59 \times 10^{-4}$ m$^4$, 0.24 m, and 2.60 m, respectively. The longitudinal resistance model of the stock rail fastener and ballast bed are simplified to bilinear models. The maximum forces associated with the longitudinal resistances of the stock rail fastener and ballast bed are 16 kN and 18 kN, respectively, and the corresponding yield displacements are all 2.0 mm [43]. The LRGRF is determined according to Table 1 and Equation (1). The joint resistance of the guardrail (JRGR) is related to the torque of the rail joint bolt, which is taken as 370 kN, and the corresponding torque is set equal to 600 N·m. In the performed calculations, it is assumed that the temperature variation amplitudes of the stock rail and guardrail are the same and equal to 40 °C.

### 3.4. Parameters of the Train Load

Herein, the typical live load of a train for the design of railway bridges is employed under the action of bending and braking conditions of the CWR according to the existing code;

see Ref. [43]. In addition, the impact coefficient of the live load has been ignored [43,44]. For mixed passenger and freight railway bridges, ZKH standard live load is adopted [45], and the wheel–rail adhesion coefficient is set equal to 0.164.

## 4. Effects of Guardrails

### 4.1. Expansion Conditions

Figure 8 illustrates the results of stock rail longitudinal force (SRLF) subjected to various installation torques of guardrail fasteners in the presence of expansion conditions, where 0 N·m indicates that no guardrail has been used. Due to the existence of guardrail joints, the guardrails expand and contract with the variation of the rail temperature, which is equivalent to the breathing section of the CWR. Hence, the temperature variation of the stock rail, guardrail, and bridges should be considered at the same time.

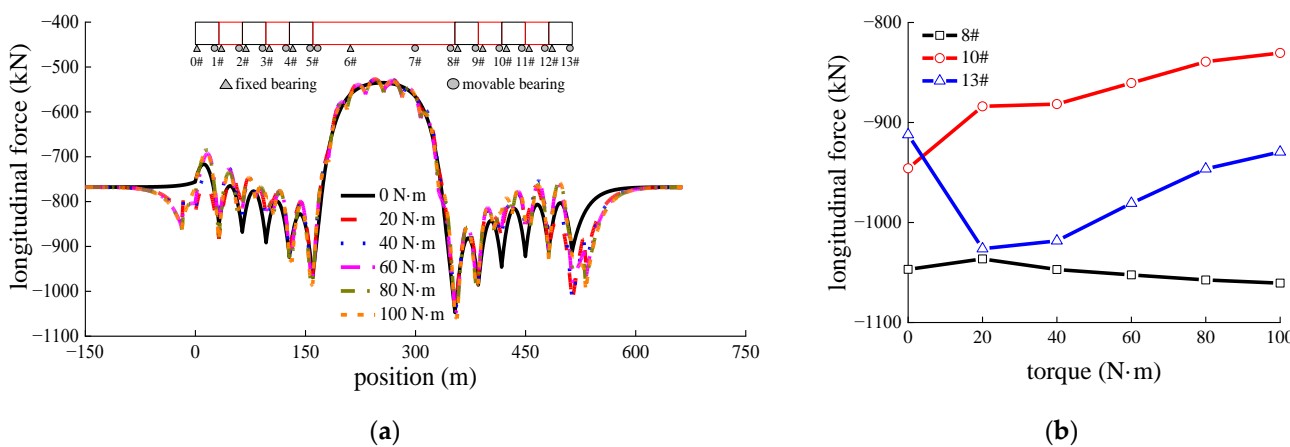

**Figure 8.** SRLF under expansion conditions. (**a**) Distribution; (**b**) relation between *M* and SRLF.

Based on the demonstrated results in Figure 8, the presence of guardrails does not have a substantial influence on the distribution rule of the SRLF, but it affects the amplitude of the SRLF. In the absence of the guardrails, the temperature span [22], which is the center distance between piers 6# and 8#, is the largest, and the maximum SRLF appears at the 8# pier position. The maximum value of SRLF is 1046.9 kN, also representing a crucial basis for both strength and stability designs of CWR on bridges. In the presence of guardrails, the SRLF at the 8# pier position slightly reduces and then gradually rises with the growth of the installation torque of the guardrail fastener. When the installation torque ranges from 20 N·m to 100 N·m, the maximum value of SRLF at the 8# pier position only increases by 24.2 kN, less than 3%. This reveals that the guardrails have little influence on the SRLF in the range of the CGB.

As the span of the simply supported bridges is close to the length of a guardrail, the SRLF in the range of simply supported bridges is remarkably influenced by the guardrails. Taking the maximum SRLF at the 10# pier position given in Figure 8a as an example, when the installation torque is placed in the range of 0–100 N·m, the SRLF lessens by 115.4 kN, indicating a reduction of 12.2%. The variation rule of the SRLF along with the installation torque at the 13# abutment position is contrary to that at the 10# pier position. By growing the installation torque to 20 N·m, the maximum SRLF at the 13# abutment position reaches 1026.3 kN, which is nearly equal to that of the 8# pier position under the same condition. For the CGB with small spans, the maximum SRLF at the abutment position is even greater than that at the end of the CGB and becomes the new basis for the design of strength and stability of CWR on bridges. However, the influence of guardrails is not considered in the design of CWR on bridges at present, and the unfavorable position of the abutment is ignored. This will increase the risk of rail fracture and lateral instability, hence reducing the safety of CWR on bridges. The existence of guardrails could also lead to variations in the longitudinal force of piers, as presented in Figure 9.

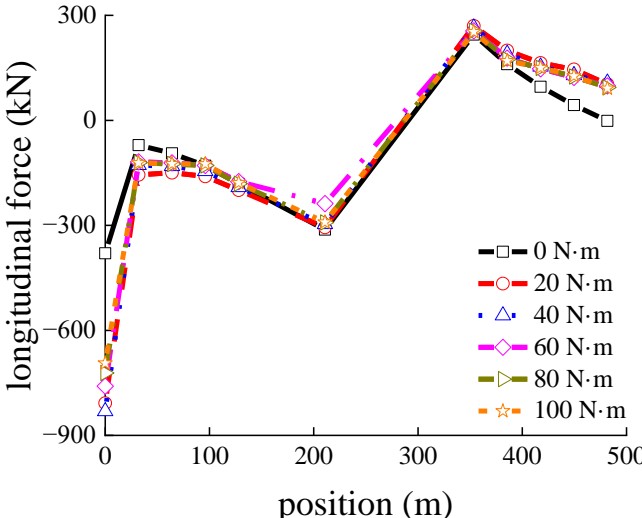

**Figure 9.** The piers' longitudinal force.

The demonstrated results in Figure 9 reveal that the piers' longitudinal force increases due to the existence of the guardrails, and the case of 0# abutment is the most significantly affected. As the installation torque rises from 0 N·m to 20 N·m, the longitudinal force of the 0# abutment magnifies by 112.8% and then lessens with the growth of the installation torque. For the case of the installation torque ranges from 20 N·m to 100 N·m, the longitudinal force of the 0# abutment decreases from 808.0 kN to 693.4 kN (i.e., a 14.1% decrease). The reason for the phenomenon is that the length of the breathing section near the 0# abutment decreases with the increase in the installation torque of the guardrail fastener, which can be seen from the distribution of the guardrail longitudinal force in Figure 10.

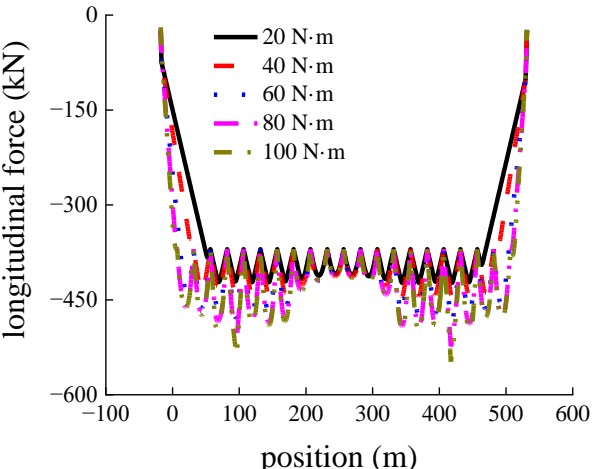

**Figure 10.** Distribution of the guardrail longitudinal force.

JRGR is also an important factor affecting guardrail deformation. The installation torque of the guardrail fastener is 60 N·m. JRGR ranges from 250 kN to 490 kN, and the interval is 60 kN. Figure 11 shows the relation between the stock rail, the longitudinal force of the 0# abutment, and the JRGR.

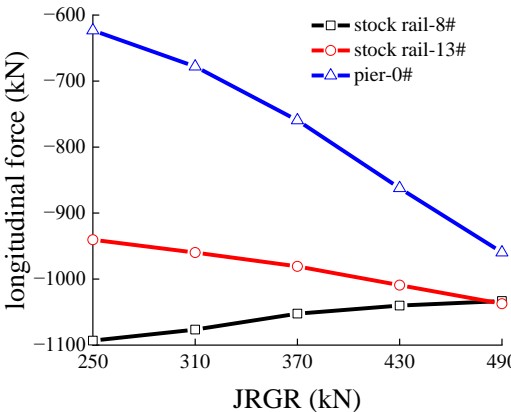

**Figure 11.** Relation between longitudinal force of stock rail, pier, and JRGR.

Figure 11 shows that the SRLF at the 13# abutment position and the longitudinal force of the 0# abutment increase approximately linearly with the increase in JRGR. The reason is that because of the increase in JRGR, the change of the guardrail gap lessens, and the guardrail deformation to the left and right abutments grows, as demonstrated in Figure 12. Such a fact exacerbates the track–bridge interaction, which causes the accumulation of the longitudinal force. The SRLF at the 8# pier position is less affected by the JRGR, and it only reduces by 59.9 kN, a decrease of 5.5% with the JRGR increasing from 250 kN to 490 kN. When the JRGR reaches 490 kN, the SRLF at the 13# abutment position is greater than that at the 8# pier position. Such a fact makes the stock rail at the 13# abutment position become the most unfavorable location under expansion conditions. It can be seen from Figure 12 that a small JRGR can lead to a big change in the guardrail gap, which results in the rail gap exceeding the structural joint gap or the rail gap being zero. So, it is recommended that the JRGR is 370–430 kN, that is, the bolt torque of the guardrail joint should be controlled in the range of 700–800 N·m.

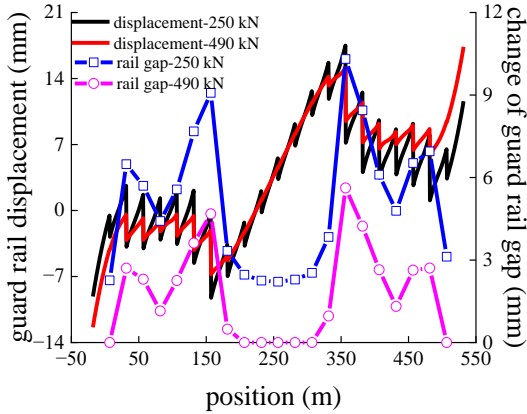

**Figure 12.** Distribution of guardrail displacement and change of guardrail gap.

### 4.2. Bending Conditions

When the train load is located at the two spans on the right side of the CGB, the double-track railway experiences the train load. The corresponding results have been presented in Figures 13 and 14. The plotted results in Figure 13 demonstrate the distribution of stock rail bending force, whereas Figure 14 illustrates the relationship between the stock rail bending force amplitude, the longitudinal force of 6# pier, and the installation torque of the guardrail fastener.

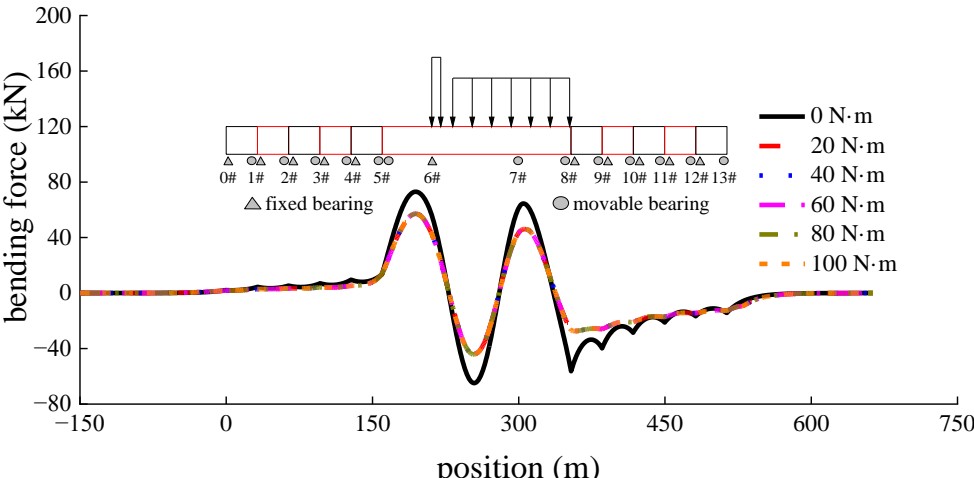

**Figure 13.** Distribution of stock rail bending force.

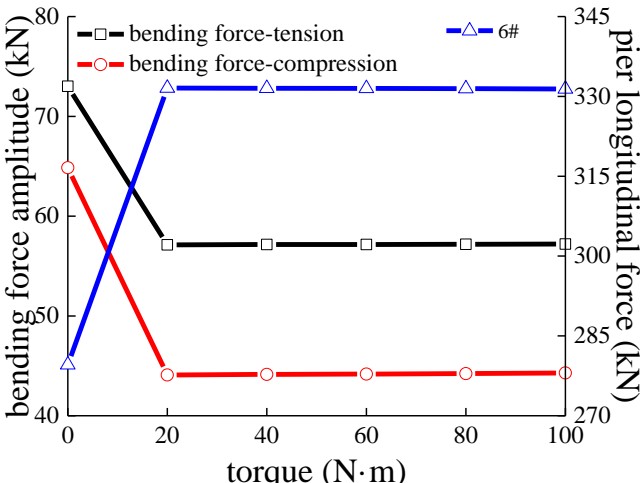

**Figure 14.** Relation between bending force amplitude, pier longitudinal force, and *M*.

As shown in Figures 13 and 14, the presence of guardrails leads to a remarkable reduction in the stock rail bending force because of the interaction between the guardrail and the stock rail. By ignoring the effect of the guardrails, the stock rail bending tension and compression force amplitudes are 73.0 kN and 64.9 kN, respectively. In the presence of guardrails as well as a 20 N·m installation torque of the guardrail fastener, the stock rail bending tension and compression force amplitudes reduce by 57.1 kN and 44.1 kN compared to the condition without guardrails. With further growing the installation torque of the guardrail fastener, the stock rail bending force presents no apparent change. The main reason is the existence of a small difference in the load–displacement curve of the LRGRF with a trivial displacement, and the bending force amplitude is less than the JRGR. Due to the train load laid on the CGB, the 6# pier undertakes the maximum longitudinal force compared with other piers and abutments, and the existence of guardrails causes the force to grow by 52.0 kN, showing an increase of 18.6%.

Based on the analysis explained above, the variation of JRGR does not have a significant effect on the stock rail bending force and pier longitudinal force, as shown in Figure 15.

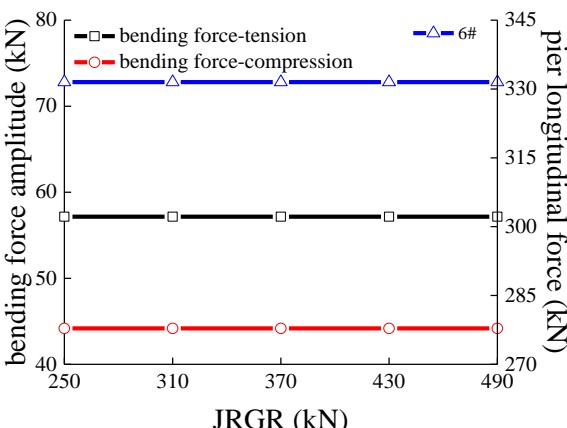

**Figure 15.** Relation between bending force amplitude, pier longitudinal force, and JRGR.

It can be seen that the presence of a guardrail could reduce the stock rail bending force and increase the pier longitudinal force, but the variable is not affected by the installation torque of the guardrail fastener and JRGR.

### 4.3. Braking Conditions

The distributions of the stock rail braking force under the action of two load arrangements are demonstrated in Figure 16. For such a loading scenario, the double-track railway load was applied. In Figure 16, the RDASB represents the relative displacement amplitude between the sleeper and the bridges.

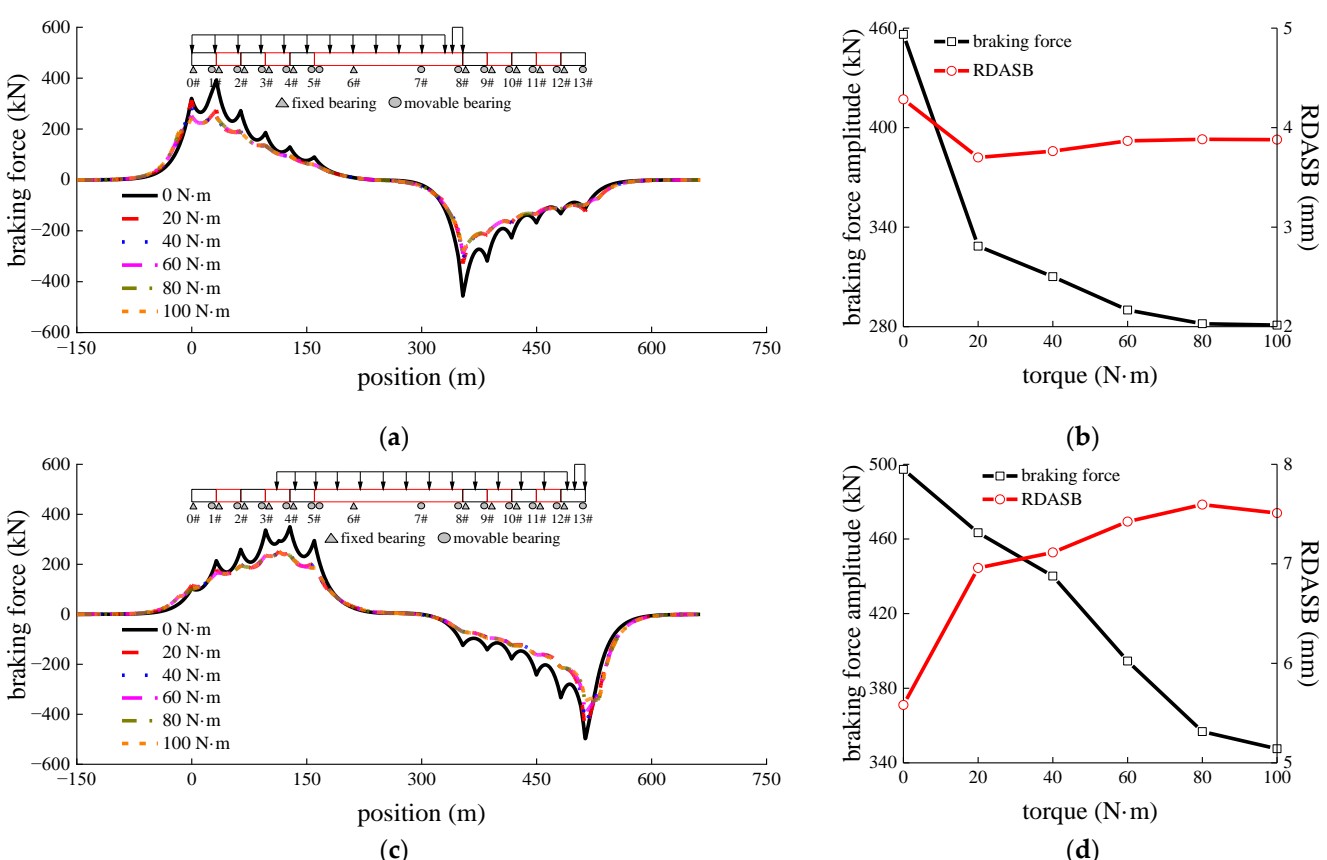

**Figure 16.** Results of braking conditions. (**a**) Distribution of stock rail braking force (load I); (**b**) relation between force amplitude, RDASB, and *M* (load I); (**c**) distribution of stock rail braking force (load II); (**d**) relation between braking force amplitude, RDASB, and *M* (load II).

The results in Figure 16 clearly indicate that the guardrails have no effect on the distribution rules of stock rail braking force subjected to two loading scenarios, but the stock rail braking force decreases due to the interaction between the guardrail and the stock rail. Taking the load I scenario as an example, the stock rail braking force amplitude in the presence of 20 N·m installation torque lessens by 127.7 kN, presenting a reduction of 28.0% compared to that without guardrails. It can be seen that the design method of the CWR on bridges in the absence of the guardrail effect makes a small allowable range of rail temperature dropping; this issue confidently limits the application of CWR technology in moderately cold regions. Moreover, when the installation torque of the guardrail fastener increases from 20 N·m to 100 N·m, the stock rail braking force amplitudes under both load scenarios lessen, respectively, by 47.6 kN and 115.7 kN (i.e., a drop of 14.5% and 25.0%). Such a fact reveals that the stock rail braking force amplitude of the load II scenario is more noticeably influenced by the installation torque of the guardrail fastener. In fact, the load of the load II scenario causes more guardrail fasteners to arrive at the yield state, and the guardrails undertake a greater longitudinal force, causing the stock rail braking force to decrease.

Comparing the results of Figure 16b,d, the RDASB of the load I scenario trivially reduces in the presence of the guardrail, that is, because the guardrail has a tendency to enhance the longitudinal stiffness of the track frame. Additionally, the small difference in the load–displacement curves of the LRGRF with small displacement subjected to various installation torques does not change the RDASB. Under the load II scenario, with the growth of the installation torque of the guardrail fastener, the RDASB exhibits a slightly increasing trend because of the guardrails laid on the subgrade, which is close to the train load.

Under braking conditions, the guardrail longitudinal force amplitude is less than the JRGR, so the change of JRGR will not significantly affect the stock rail braking force and RDASB, as demonstrated in Figure 17. For the plotted results, the installation torque of the guardrail fastener in the calculations is 60 N·m.

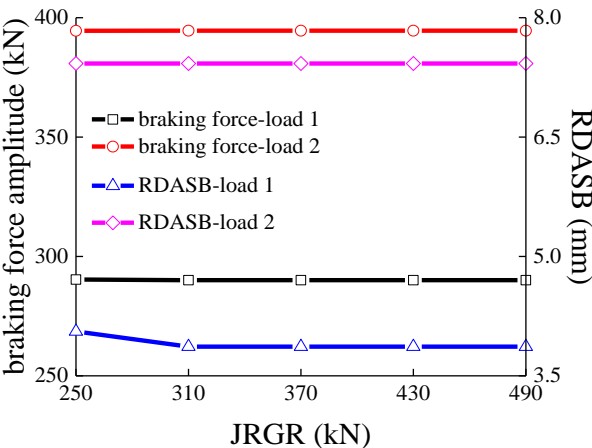

**Figure 17.** Relations among the amplitude of braking force, the RDASB amplitude, and JRGR.

### 4.4. Rail Breaking Conditions

According to the results displayed above, an additional SRLF peak occurs at the 13# abutment position when taking into account the guardrails. As the values of SRLF at the 8# pier position and 13# abutment position are almost equal, the locations of the rail breaking are set at the two positions for two calculation conditions. Figure 18 demonstrates the results of the rail breaking conditions.

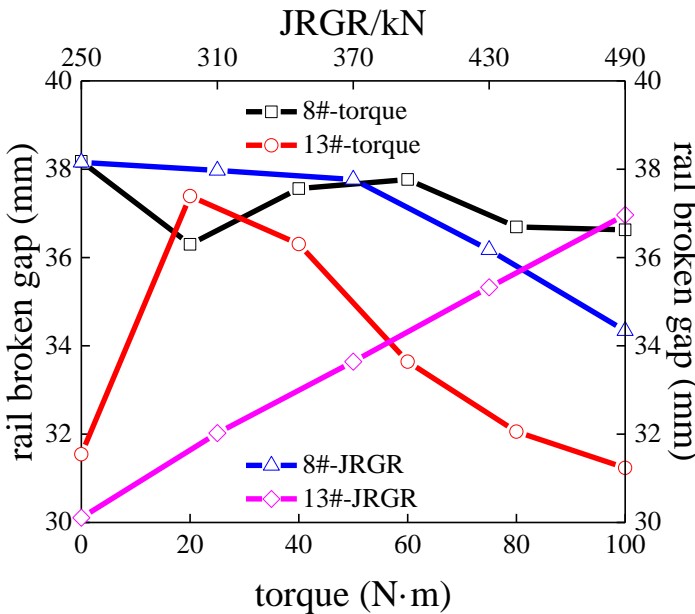

**Figure 18.** Relations among *M*, JRGRB, and rail broken gap.

The essence of the rail broken gap is the release of SRLF under expansion conditions. By this virtue, a close comparison between the plotted results in Figure 16 and those of Figures 8b and 9 reveals that the change rule of the rail broken gap in terms of the change of installation torques of the guardrail fastener and JRGR is consistent with that of the SRLF amplitude under expansion conditions. From a numerical aspect, the rail broken gaps with different calculation parameters are all lower than the limits set in the code [43], and the change of the rail broken gap due to the calculation parameters is trivial. Hence, under rail breaking conditions, it has little effect on the design of the CWR on bridges with or without guardrails.

## 5. Maximum Longitudinal Stiffness of Piers for CGBs

To ensure that the design method of the CWR on bridges in the absence of guardrails meets the actual needs, the maximum longitudinal stiffness of the pier for the CGB is determined according to the equal-strength design principle. By taking into account the guardrails effect, the maximum SRLFs at the right abutment position under expansion and braking conditions in order are denoted by $F_{hs}$ and $F_{hz}$ (the longitudinal stiffness of piers for the simply supported bridge is taken according to the minimum value specified in the code [43]). In the absence of the guardrails effect, the maximum SRLFs under expansion and braking conditions are denoted by $F_s$ and $F_z$, respectively. Based on the design method of stability and strength of the CWR on bridges [43], the following requirements should be met:

$$F_s \geq F_{hs}, \tag{2}$$

$$F_s + F_z \geq F_{hs} + F_{hz}, \tag{3}$$

According to the influence of the installation torque of the guardrail fastener and JRGR on the SRLF under expansion and braking conditions, the installation torque and JRGR are taken as 40 N·m and 370 kN for further analysis. In addition, the relationship between $F_{hs}$, $F_{hz}$, $F_s$, and $F_z$ and the longitudinal stiffness of #6 pier with fixed bearings is presented in Figure 19.

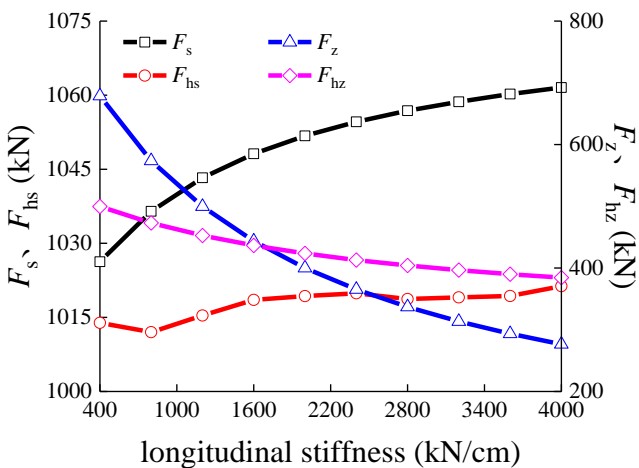

**Figure 19.** Effect of the longitudinal stiffness of #6 pier.

Figure 19 displays that in order to meet Equations (2) and (3), the maximum longitudinal stiffness of the pier with fixed bearings for the CGB would be 2147.3 kN/cm. Based on the proposed method, the maximum longitudinal stiffness of the piers associated with the fixed bearings is evaluated for the common span CGBs in the case of mixed passenger and freight railways. A summary of these calculations is presented in Table 3.

**Table 3.** Predicted results of the longitudinal stiffness for piers.

| Spans of CGB (N·m) | Lower Limit of Piers Based on Equation (2) (kN/cm) Rail Temperature Variation | | | Upper Limit of Piers Based on Equation (3) (kN/cm) Rail Temperature Variation | | | Recommended Maximum Value (kN/cm) |
|---|---|---|---|---|---|---|---|
| | **30 °C** | **40 °C** | **50 °C** | **30 °C** | **40 °C** | **50 °C** | |
| 32 + 48 + 32 | - | - | - | 163.9 | 129.9 | 101.6 | - |
| 40 + 64 + 40 | - | - | - | 652.0 | 617.3 | 612.2 | - |
| 44 + 80 + 44 | 1177.9 | 1338.9 | 1605.8 | 1279.1 | 1245.1 | 1281.6 | - |
| 50 + 92 + 50 | 400.0 | 400.0 | 400.0 | 2214.2 | 2147.3 | 2122.7 | 2120 |
| 60 + 100 + 60 | 400.0 | 400.0 | 400.0 | 3207.2 | 3033.3 | 2968.0 | 2960 |

"-" indicates that no pier stiffness exists that meets the requirements.

As presented in Table 3, when the temperature span of the CGB is small, the SRLF at the abutment position accounting for the guardrail effect is always greater than the SRLF amplitude without a guardrail under expansion conditions. In such a case, the guardrail must be considered in the CWR design. For the case of large spans of the CGB, Equation (2) is always satisfied, and the maximum longitudinal stiffness pier could be essentially evaluated by Equation (3). With the growth of the spans of CGB, the upper limit value of the longitudinal stiffness of piers grows, and the recommended maximum longitudinal stiffness of piers for (50 + 92 + 50) m and (60 + 100 + 60) m CGBs would be 2120 kN/cm and 2960 kN/cm, respectively. If the longitudinal stiffness of the pier is greater than the recommended maximum value, the influence of the guardrail must also be considered in the design of the CWR on the bridge.

## 6. Conclusions

In this study, load–displacement curves and mathematical models representing the LRGRF are obtained from the longitudinal resistance test of the guardrail fastener. These plots are then effectively introduced to the stock rail-guardrail-sleeper-bridge-pier integrated simulation model to examine the influences of the LRGRF and JRGR on track–bridge interaction. The main results obtained are summarized as follows:

(1)　The Langmuir EXT1 function is employed as an effective mathematical model for the rising stage of the load–displacement curve that represents the LRGRF. The nominal

longitudinal resistance and the elastic longitudinal displacement of the rail prior to slipping grow approximately linearly with the increase in the installation torque. As the installation torque varies from 40 N·m to 60 N·m, the nominal longitudinal resistance and longitudinal displacement of the rail prior to slipping change from 3.91 kN to 6.46 kN and from 0.15 mm to 0.29 mm, respectively.

(2)  The interaction between the stock rail and the guardrail could lessen the track–bridge interaction of the CWR on the bridge. Additionally, this results in a reduction of both the SRLF and the rail broke gap, with a stock rail braking force reduction of up to 28.0%.

(3)  The existence of a breathing zone of the guardrail can exacerbate the track–bridge interaction of the CWR near the abutment with moveable bearings. In a special case, when the spans of the CGB and the longitudinal stiffness of piers for simply supported bridges are small, the SRLF at the abutment position is even greater than the force at the CGB end position. This enables the abutment position to become a new control point for the design of CWR on bridges.

(4)   If the installation torque of the guardrail fastener is too high, the SRLF under expansion conditions greatly increases. Conversely, if the torque is too small, the stock rail braking force noticeably grows. Hence, in general, an installation torque of 40–60 N·m is recommended. If the JRGR is too large, the SRLF under expansion conditions also increases significantly. If the JRGR is too small, the variation of the guardrail gap is large. Therefore, the JRGR is generally recommended to be in the interval of 370–430 kN.

(5)  If the temperature span of the CGB is lower than 124 m, the influence of guardrails must be considered in the design of the CWR on bridges. When the longitudinal stiffness of the pier for the CGB is less than the recommended maximum value proposed in the paper, the existing simplified model without guardrails can be effectively employed in the design of the CWR.

**Author Contributions:** Conceptualization, K.X.; data curation, W.D.; funding acquisition, K.X. and W.Z.; investigation, K.X. and W.D.; methodology, K.X. and H.X.; writing—original draft preparation, K.X.; writing—review and editing, W.D., H.X. and W.Z. All authors have read and agreed to the published version of the manuscript.

**Funding:** This research was funded by the National Natural Science Foundation of China, grant numbers 52008272 and 51978423. This research was also funded by the Natural Science Foundation of Hebei Province, grant numbers E2022210046 and E2021210099.

**Institutional Review Board Statement:** Not applicable.

**Informed Consent Statement:** Not applicable.

**Data Availability Statement:** All data and models used in the study appear in the submitted manuscript.

**Conflicts of Interest:** The authors declare no conflict of interest.

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
