# Peer review of "Influence of Guardrails on Track–Bridge Interaction with a Longitudinal Resistance Test of the Fastener"

_applsci, doi:10.3390/app13063750_

Round 1
Reviewer 1 Report
Review
The paper presents a very interesting topic entitled “Influence of Guardrail on Track-Bridge Interaction with Longitudinal Resistance Test of the Fastener”. The paper can be accepted for the possible publication after the minor revision. The following comments should be addressed:
0. Overall. There are a few grammatical mistakes in the paper which need to be improved.
1. Line: 149. To strengthen the introduction section of the paper, the authors are suggested to cite the following journal stating the bending deformation of the steel plate with respect to neutral axis under cyclic loading:
Saleem, A., Tamura, H., & Katsuchi, H. (2022). Change in Surface Topography of Structural Steel Under Cyclic Plastic Deformation. Proceedings of the 9th International Conference on Fracture, Fatigue and Wear, 175–193. https://doi.org/10.1007/978-981-16-8810-2_14
2. Line: 158. and this paper have been presented in Table 2 à and the results from the reference paper have been presented in Table 2.
3. Line: 199. Why the impact coefficient of live load has been ignored in the design of railway bridge? Please also refer to the following paper wherein the dynamic load allowance was ignored for long-span bridges because of their flexibility and capability to easily absorb impact. What similarities or dissimilarities authors can draw between the example railway bridge and the one shown in the reference paper.
Ali, K., Javed, A., Mustafa, A. E., & Saleem, A. (2022). Blast-Loading Effects on Structural Redundancy of Long-Span Suspension Bridge Using a Simplified Approach. Practice Periodical on Structural Design and Construction, 27(3). https://doi.org/10.1061/(asce)sc.1943-5576.0000699
4. Line: 412. Additionally, this issue results in à this results in

Reviewer 2 Report
The revised paper has clear description logic and high reference value, and the work is basically full. However, there are still some problems that need to be revised:
1. The abbreviation of continuous welded rail, CWR, has appeared for the first time in the abstract, so it does not need to be explained again in the introduction.
2. In the introduction, references 28 and 29 should be cited in the same format as other references, that is, directly after the sentence, not in the form of the superscript.
3. The introduction contains some old references and many sentences without citations. It’s good to cite these very recent and related papers in the introduction and to support your discussion referring to these papers.
Ø Track–Bridge Interaction of CWR on Chinese Large-Span Bridge of High-Speed Railway, doi: https://doi.org/10.3390/app12189100.
Ø Dynamic Response of Spatial Train-Track-Bridge Interaction System Due to Unsupported Track Using Virtual Work Principle, doi: https://doi.org/10.3390/app12126156.
Ø A novel prediction model of packing density for single and hybrid steel fiber-aggregate mixtures, doi: https://10.1016/j.powtec.2023.118295.
4. Figure 3: The fitted curve should be implemented with a brighter color and a thicker width than other curves; the curve for the 2th test could be a centerline.
5. Lines 115-116, 119-120, 124-126: Although Applied Sciences journal has rule of “All Figures, Schemes and Tables should be inserted into the main text close to their first citation and must be numbered following their number of appearance (Figure 1, Scheme I, Figure 2, Scheme II, Table 1, etc.).”, the reviewer still suggests that these single sentences be written in a complete paragraph, which does not conflict with the above rule.
6. Figure 7 and the other remaining figures: The width of curves could be larger.
Reviewer 3 Report
The most important achievements:
(1) The test results provide parameters for the standardization design of continuous welded rail on bridges.
(2) The integrated simulation model and the recommended maximum value of continuous girder bridges can simplify the design process of continuous welded rail, and the suggestions about the installation torque of the guardrail fastener and guard- rail joint can provide guidance for the maintenance of continuous welded rail track.
(3) The results give new basis and reference for improving the industry specifications for design and maintenance of railway continuous welded rail.
After proofreading from English, you can publish the paper
